mental health; mobile application; Nepal; mhGAP; app; guideline

**Corresponding author:**
Nagendra P. Luitel;
Email: nagendra.luitel@email.gwu.edu

# Development and functioning of the mobile app-based mh-GAP intervention guide in detection and treatment of people with mental health conditions in primary healthcare settings in Nepal

Nagendra P. Luitel[1,2] ⬤, Kriti Pudasaini[2], Pooja Pokhrel[2], Bishnu Lamichhane[2], Kamal Gautam[1,2] ⬤, Sandarba Adhikari[2], Akerke Makhmud[3], Tatiana Taylor Salisbury[3], Nicole Votruba[3,4,5] ⬤, Eric Green[6], Neerja Chowdhary[7], Mark J.D. Jordans[2,3] ⬤, Brandon A. Kohrt[1,2] ⬤, Tarun Dua[7], Graham Thornicroft[3,8] and Kenneth Carswell[7]

[1]Center for Global Mental Health Equity, Department of Psychiatry and Behavioral Health, George Washington University, Washington, DC, USA; [2]Research Department, Transcultural Psychosocial Organization (TPO) Nepal, Baluwatar, Nepal; [3]Centre for Global Mental Health, Health Service and Population Research Department, Institute of Psychology, Psychiatry & Neuroscience, King's College London, London, UK; [4]Nuffield Department of Women's & Reproductive Health, University of Oxford, Oxford, UK; [5]The George Institute for Global Health UK, Imperial College London, London, UK; [6]Duke Global Health Institute, Duke University, Durham, NC, USA; [7]Department of Mental Health and Substance Use, World Health Organization, Geneva, Switzerland and [8]Centre for Implementation Science, Health Service and Population Research Department, Institute of Psychology, Psychiatry & Neuroscience, King's College London, London, UK

## Abstract

This paper describes the development process of a mobile app-based version of the World Health Organization mental health Gap Action Programme Intervention Guide, testing of the app prototypes, and its functionality in the assessment and management of people with mental health conditions in Nepal. Health workers' perception of feasibility and acceptability of using mobile technology in mental health care was assessed during the inspiration phase ($N = 43$); the ideation phase involved the creation of prototypes; and prototype testing was conducted over multiple rounds with 15 healthcare providers. The app provides provisional diagnoses and treatment options based on reported symptoms. Participants found the app prototype useful in reminding them of the process of assessment and management of mental disorders. Some challenges were noted, these included a slow app prototype with multiple technical problems, including difficulty in navigating 'yes'/'no' options, and there were challenges reviewing detailed symptoms of a particular disorder using a "more information" icon. The initial feasibility work suggests that if the technical issues are addressed, the e-mhGAP warrants further research to understand if it is a useful method in improving the detection of people with mental health conditions and initiation of evidence-based treatment in primary healthcare facilities.

## Impact statement

The task-sharing approach has been widely advocated to reduce the treatment gap for mental health care in low- and middle-income countries (LMICs) where mental health professionals are scarce. Mobile technology can be of assistance to help reduce some of the barriers to implementing this. Through collaboration with primary healthcare providers in Nepal and Nigeria, we co-designed a mobile app version of the World Health Organization's mental health Gap Action Programme Intervention Guide (mhGAP-IG). The mhGAP-IG supports the detection and treatment of mental health symptoms by healthcare providers who are not mental health specialists. The app includes a decision-making function that provides a provisional diagnosis and treatment options based on patients' presenting symptoms and background information. Throughout the Emilia project, the data collected through the app was also used to provide remote supervision for primary healthcare providers in an effort to address another major barrier to implementing this task-sharing approach effectively. This study suggests that if the technical issues are addressed, the mhGAP-IG mobile app could be a potential strategy to address the low detection of mental health conditions in primary care that warrants further research and exploration.

## Introduction

Although the task-sharing approach in mental health care has widely been implemented in high-, middle- and low- income countries, the detection rate of people with mental disorders, particularly depression, by trained primary healthcare providers has been found to be low. A recent systematic review revealed that trained primary healthcare providers in low- and middle-income countries (LMIC) detected between 0% and 28% of people with depression correctly (Fekadu et al., 2022) whereas the proportion of correct detection of depression in high-income countries (HICs) was about 48% (Mitchell et al., 2009). Among those who were correctly detected with depression, only about 4% received minimally adequate treatment in LMICs and 20% in HICs (Thornicroft et al., 2017) based on the definition of minimally adequate treatment which in summary is either ≥1 month of a medication plus ≥4 visits to any type of medical doctor or ≥ 8 visits with any professional (Wang et al., 2007). Depression and other common mental disorders are highly prevalent among people attending primary healthcare facilities (Rait et al., 2009; Serrano-Blanco et al., 2010), and people with mental health conditions are more likely to contact primary healthcare providers rather than mental health specialists (Reilly et al., 2012) because of physical accessibility, acceptability of care and potentially reduced stigma when receiving mental health treatment in primary healthcare (WHO and Wonca, 2008). The low levels of detection and treatment initiation of people with mental disorders by trained primary healthcare providers pose a serious concern and threat to the scale-up of mental health services in the LMICs where specialist mental healthcare providers are limited.

In Nepal, the task-sharing approach has been widely implemented to address the scarcity of mental health services in rural areas where specialist mental health services do not exist (Chase et al., 2018; Luitel et al., 2020). Although the government has taken the initiative to scale up the WHO mhGAP-based treatment package throughout the country, our recent study revealed that trained primary healthcare providers were able to detect only 24% of depression correctly immediately after the mhGAP-based training, with almost every patient being provided minimally adequate treatment as recommended by the mhGAP protocol (Jordans et al., 2019). These results indicate that if mental disorders are correctly detected, trained healthcare providers can initiate the treatment as recommended by the mhGAP protocol. We hypothesize that the mobile application-based mhGAP guideline can help to increase the correct detection, initiation and continuation of evidence-based treatment for mental disorders in primary care. The EMILIA (e-mhGAP Intervention Guide in LMICs: proof-of-concept for Impact and Acceptability) research program consortium developed a mobile app-based clinical guideline that provides practical guidance to primary healthcare providers in detection, treatment, and follow-up of people with mental health conditions (Taylor Salisbury et al., 2021b). The app was developed in collaboration with the WHO and followed the WHO mhGAP-IG V2 protocol and guidelines (WHO, 2016).

Mobile technology is increasingly being used in mental health care, and numerous mobile phone-based applications are now available (Torous and Roberts, 2017). Previous studies on the use of digital platforms to support decision-making and diagnosis by primary healthcare providers show promising results (Maulik et al., 2017, Naslund et al., 2017). However, several issues hindering the use of digital platforms are for instance, a lack of consistent and reliable wireless internet access in the community, low speed of mobile networks, low digital literacy among primary healthcare providers, and cost-effectiveness of digital intervention (Khan et al., 2020, Naslund et al., 2017, Wallis et al., 2017). In Nepal, only a few studies have assessed if mobile application can be used in healthcare delivery, particularly in sending reminders for follow-up and assessing signs and symptoms of illness (Harsha et al., 2015); SMS reporting of neonatal health information and malaria surveillance (Shrestha, 2014); and support health workers in the rural district hospitals (Bhatta et al., 2015) but none have attempted to use a clinical decision support tool based on mhGAP in Nepal. In this paper, we present information on the development process of a mobile app-based clinical guide, on testing the app prototype, and on its functionality in the assessment and management of mental disorders in primary healthcare settings in Nepal.

## Methods

### Settings

The development and prototype testing of the mobile app was conducted in Chitwan, a relatively urban district in south-central Nepal (Gyenwali et al., 2017). Nepal has a mixed healthcare delivery system, comprising the public sector, private for-profit sector, and non-governmental organizations (NGOs) (Mahat et al., 2018). There are 0.76 psychologists, and 0.66 psychiatrists per 100,000 population in Nepal (Rai et al., 2021). As specialist mental health services are concentrated in the hospitals located in big cities, traditional healers are the primary source of mental health treatment in many rural areas (Pham et al., 2021). Primary healthcare providers in Chitwan were trained on the WHO mhGAP-IG through the PRIME project (Luitel et al., 2020).

### Study design

We used the design thinking methodology (Brown and Wyatt, 2010), an innovative human-centered design (HCD) approach (IDEO, 2015), for the development and prototype testing of the mobile app-based clinical guideline. Design thinking is a pragmatic and experiential methodology that helps to address the needs of the people who consume the product or service (Brown and Wyatt, 2010). This approach brings end-users and developers together to co-create interventions and delivery strategies (Bazzano et al., 2017). This method has also been used in the development of a range of user-friendly mental health interventions, for example, a WHO transdiagnostic chatbot for distressed youth (Hall et al., 2022), evidence-based psychotherapies in low-resource settings (Lyon and Koerner, 2016), and mental health support for adolescent mothers in Mozambique (Taylor Salisbury et al., 2021a).

We involved primary healthcare providers who were trained on WHO mhGAP-IG V2 throughout the study. The app prototype development process comprises of three distinct steps summarized below.

### Inspiration phase

The inspiration phase of the study was conducted with 43 purposively selected mhGAP-IG-trained healthcare providers using semi-structured interviews and focus group discussions. This phase focused primarily on understanding healthcare providers' needs, use of smartphones in their day-to-day life, and prior experiences of using mobile phones in healthcare delivery. We also assessed

healthcare providers' perception of feasibility, acceptability, and usefulness of app-based clinical guides on mental health care and possible barriers. The details of the inspiration phase study can be found elsewhere (Pokhrel et al., 2021). The main objectives of this phase were to understand the technology use and needs of possible end users and potential challenges with implementing digital health.

### Ideation phase

The mobile app-based prototype was led by WHO and members of the EMILIA consortium and a software development company contracted for the process, which included digital programmers and designers. The contents and functions of the prototype were determined based on the results of the inspiration phase, and consultation with mental health experts and supervisors. The members of the research team who were involved in the inspiration phase were also involved in creating the prototype. The objective of this phase was to create a prototype for subsequent testing.

### Implementation phase

Implementation of the prototype (i.e., prototype testing) was conducted in three iterative rounds with 15 purposively selected healthcare providers. In Round I, participants ($N = 8$) were asked to perform a role-play (i.e., assessment of a possible depression case using the app prototype), and their experience was collected through semi-structured interviews. The lessons and feedback provided in Round I were incorporated in the prototype before Round-II prototype testing, which followed the same procedure. Review of the revised prototype, role play using a standardized vignette, and semi-structured interviews ($N = 7$) were the methods used in Round II. The final round of the prototype testing ($N = 9$) focused more on information to be collected using the app during assessment, management, and follow-up care. Some healthcare providers participated in multiple rounds of prototype testing, hence the total number add-up over 15. The aim of all phases was primarily to test the technical and functional aspects of the prototype in an iterative fashion.

### Interview process

Two qualitative researchers conducted the inspiration phase interviews and focus group discussions (FGDs), with one facilitating and the other notetaking. The FGDs were audio-recorded and took approximately 60 to 90 min (Pokhrel et al., 2021). A female actor was involved in the prototype testing for the role plays. All activities took place in the respective health facilities with informed consent from each health worker involved.

The Round-I prototype testing was conducted in two parts. First, healthcare providers were asked to perform an assessment and diagnosis of mental health problems based on a standard roleplay without using the prototype. After this, healthcare providers were briefly oriented to the prototype and asked to perform another assessment with the same role play, but this time using the prototype. After the second role-play, participants were asked to share their experience of using the prototype in assessment, including additional features to be included in the prototype that can help with assessment and management. The prototype was adapted before Round II prototype testing. In Round II the same prototype-based review and interview process was used. The participants who were not involved in Round I prototype testing were

briefly oriented on the prototype and took part in a roleplay using the same procedure as Round I. Their experience, feedback, and comments were collected through semi-structured interviews. Round III of the prototype testing focused on follow-up care and patients' data using the same process as above. Afterward, participants were asked about the information that was required in follow-up care.

### Data analysis

We used the content analysis method to analyze the qualitative data collected from the prototype testing (observation) and subsequent follow-up interviews with healthcare providers. The content analysis method allows data coding from both deductive and inductive approaches (Bengtsson, 2016). The interview guides and observation checklists used in the study were short and in a semi-structured format. Therefore, the data collectors summarized the challenges and difficulties of using the prototypes, and the recommendations made by the participants in a summary form instead of transcribing the interviews in detail. Themes were generated by reviewing all the collected data by the first and second authors. The initial themes and concepts generated were discussed with other members of the research team. Data were analyzed and interpreted manually using the summary notes created by the data collectors.

## Results

The prototype testing was conducted with 15 primary healthcare providers who had between 7 months to 9 years of experience in using the WHO mhGAP Intervention Guide. The age of the participants ranged from 25 to 46 years with a mean age of 35.3 years. Of the total, 13 participants were male, 2 MBBS doctors (Bachelor of Medicine and Bachelor of Surgery), 9 HA (Health Assistant) and 4 AHW (Auxiliary Health Worker). The results are presented separately by different phases of the study: inspiration phase, ideation phase and implementation or prototype testing phase.

### Inspiration phase

The inspiration phase identified that most of the healthcare providers had a smartphone, and they used smartphones for phone calls, viewing social media, and calling friends through messenger applications. Although many healthcare providers did not have experience of using a mobile phone application in health care, they were highly motivated and positive about this in mental health care. Suggested diagnosis and treatment options, messaging to promote follow-up care, minimal typing option in the application, increasing speed and offline function were the commonly recommended features and functions for the mobile app. Healthcare providers believed that a mobile app could help bring uniformity in diagnosis and management of mental health conditions across primary healthcare facilities. The details of the inspiration phase study can be found elsewhere (Pokhrel et al., 2021).

### Ideation phase

The app prototype was developed based on feedback and recommendations made by healthcare providers in the inspiration phase qualitative study, and consultation with mental health experts, trainers and supervisors. As recommended by the healthcare

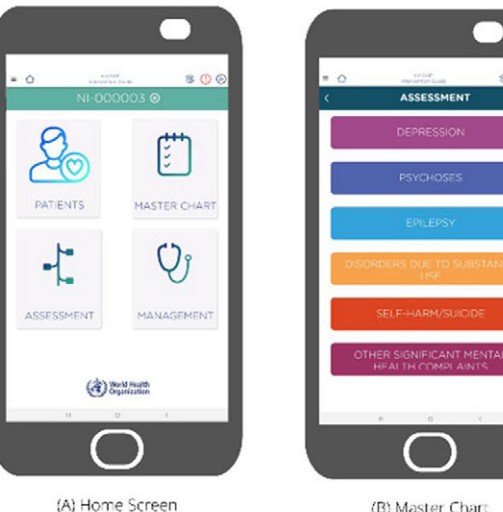
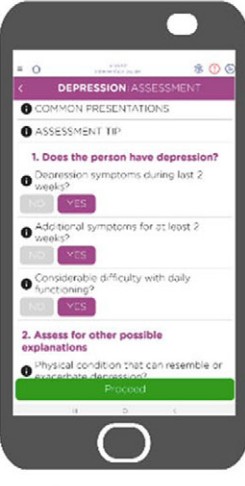
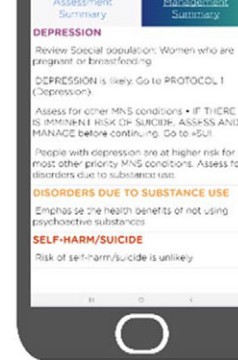

{A} Home Screen          {B} Master Chart          {C} Depression assessment          {D} Assessment summary

**Figure 1.** Assessment process in the app prototype.

providers, the content of the prototype was developed in both Nepali (local language) and English language. The prototype included a feature that provided provisional mental health condition and management options based on the reported symptoms and characteristics of people (Figure 1). The prototype comprised of four modules on the home page which included a Master Chart, Module, Management, and Medicine.

The prototype used as its reference information mhGAP-IG v2 (WHO, 2016). All information in the prototype was consistent with this.

### Master chart

The master chart module contains the core symptoms of a particular mental health condition that can be used by healthcare providers during learning of assessment and management. In the prototype, when a particular condition is selected, it automatically links to the "Module" section (see below). Although the initial plan was to develop a mobile application including six WHO mhGAP priority mental health conditions (depression, psychosis, epilepsy, disorders due to substance use, self-harm/suicide and other significant mental health complaints), the draft prototype included depression, self-harm/suicide and disorders due to substance use only.

### Management

The management module can be used after assessment is completed and provides core information for managing mental health conditions. When a possible mental health condition is identified, the prototype automatically takes the user to the relevant condition's management protocol. Once a relevant protocol is selected, it provides a list of clinical management options.

### Medication

The 'medication' module contains details of medicines that can be used for management of a particular mental health condition. When a mental health condition is detected, the prototype routinely provides a list of relevant medicines that can be prescribed along with reference to adaptations to make for special populations, for example, child/adolescent and older adults.

### Module

The 'module' is used for assessment of the person with a possible mental health condition. When healthcare providers suspect a particular condition, they can choose that condition in the 'Module' option for further assessment questions. The Module contains a list of major symptoms of a particular condition with 'yes' or 'no' answers. Additional information from the simplified screen can be accessed through selection of the 'i' icon.

### Implementation phase (prototype testing)

Prototype testing was conducted in three distinct and iterative rounds. Healthcare providers who were trained on WHO mhGAP intervention guide were involved in both the inspiration and prototype testing phase of the study. Additionally, a few healthcare providers who were not involved in the inspiration phase were also included in the prototype testing. Results from each round of prototype testing have been presented separately.

### Round-I

Healthcare providers reported that the prototype was slow, and the extra information icon (the 'i' icon) took a long time to open and disappear. Multiple additional mobile devices/tablets were provided during role plays because devices often froze (technical issues). This could be the reason only a few healthcare providers ($n = 4$) used the 'i' icon during the role plays. The same problem of speed was also reported while using the 'yes' / 'no' buttons and these did not work so well ($n = 4$). While there were a lot of problems in the 'yes' and 'no' options, these functions were one of the most useful features reported by the healthcare providers. Some healthcare providers ($n = 2$) reported that the prototype provided different diagnoses than what they had suspected. The discrepancy in diagnosis could be because of several reasons, such as technical problems in the mobile application (e.g., freezing of app, *etc.*) and competencies of the healthcare providers in accurately diagnosing the conditions.

As healthcare providers were overly engaged in the prototype during the assessment (e.g., reading and selecting symptoms in the prototype), their interactions with the person with the mental health condition was sometimes disconnected ($n = 5$). This may

explain why two healthcare providers stopped using the prototype during the role play. Despite multiple technical and practical problems in the draft prototype, most healthcare providers were positive and motivated using the prototype. They reported that the prototype reminded them of key aspects to be assessed. Among those healthcare providers who did not face any technical problems, they completed the assessment as per WHO mhGAP-IG V2. Some information that was duplicated in the Master Chart was recommended for removal (e.g., common presentation of depression). Healthcare providers recommended adding a PDF of the mhGAP-IG manual, and videos tutorials related to the mhGAP-IG in the prototype. As the draft prototype did not include the content of psychological intervention, healthcare providers also suggested adding detailed content of psychological interventions in the prototype.

### Round-II

The prototype had undergone multiple revisions based on the Round-I feedback from healthcare providers. A major feedback issue in Round I was the addition of an electronic record for each person with a possible mental health condition to monitor contacts. To allow for this, a registration and log-in page was added to protect confidentiality of individual data and was accessed under a new module "patients". Due to text character limitations on the app home screen and based on feedback from end users about ease of understanding, this was called 'Patients'. The contents and functions included under the "Medication" were transferred to 'Management' in the revised prototype. The 'Patients' module allows healthcare providers to create a basic electronic record for each person with a possible mental health condition with key demographic information; however, this function was not active in the Round-II prototype testing. Another change made in the prototype was in the layout of the 'Master Chart'. In the previous version, the 'Master Chart' included a list of all mhGAP disorders along with their common symptoms whereas in the revised prototype, a 'show/hide' option was added where healthcare providers can select a particular disorder they want to review. The Round-II prototype testing started with a review of the revised prototype to ensure that recommended changes were incorporated in the revision. Most of the healthcare providers

($n = 6$) liked the log-in option which helped in protecting personal information. Most healthcare providers ($n = 5$) recommended changing the layout of the front page with 'Patients' module at the beginning followed by 'Master Chart', 'Module' and 'Management' ($n = 5$). Some healthcare providers ($n = 5$) recommended a summary of both assessment and management at the end of the management section to verify if their assessment and management are consistent with the results provided by the prototype.

Healthcare providers can initiate assessment using both 'Master Chart' and 'Patient' modules (Figure 2). Most healthcare providers ($n = 6$) preferred assessment through 'Patients' module. A few healthcare providers ($n = 2$) recommended separating psychological and pharmacological treatments by adding a function in the prototype where the application should not proceed to pharmacological treatment until psychological interventions are provided. Other recommendations made by the participants were collecting 'completed years of age' instead of 'date of birth' ($n = 3$) and details of drugs interactions in the management section. Most of the participants ($n = 5$) recommended the content of the prototype in both Nepali and English languages.

### Round-III

The aim of the final round of the prototype testing was to collect patients' information for follow-up care. All healthcare providers ($n = 9$) suggested that individual's name, age, sex, occupation, and marital status are important and necessary information for assessment and management of mental health conditions in Nepal. They also suggested including place of residence and phone number ($n = 6$) to make follow-up care easier. On top of the contact details such as municipality, wards, and village name, participants ($n = 7$) also suggested including caregivers' names and contact numbers ($n = 6$) in the application. They also suggested including information about history of mental illness in the family ($n = 6$). Healthcare providers considered follow-up care as an important aspect in mental health treatment, therefore, they recommended assessing whether patients are receiving recommended doses of the medicines ($n = 6$) and psychological intervention, side effects of psychotropic medicines ($n = 6$), improvement in the patients' health condition ($n = 4$), and

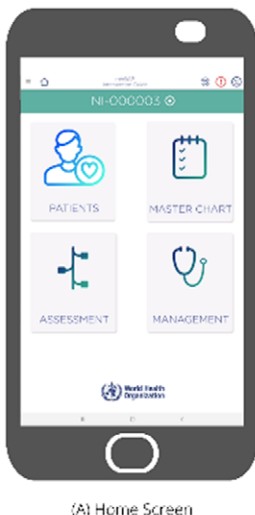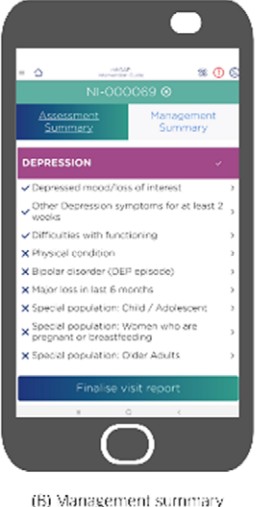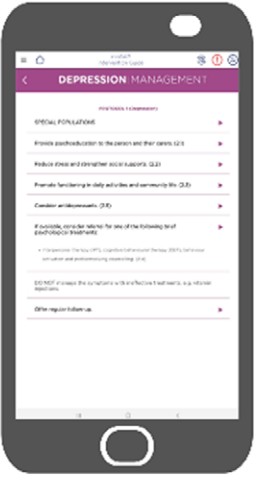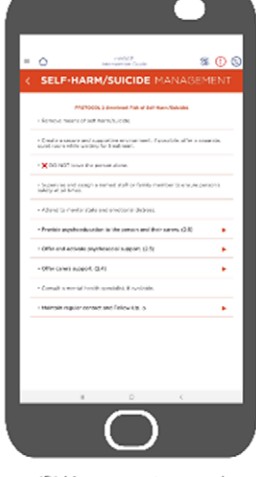

(A) Home Screen    (B) Management summary    (C) Management protocol    (D) Management protocol

**Figure 2.** Management process of a mental health condition in the app prototype.

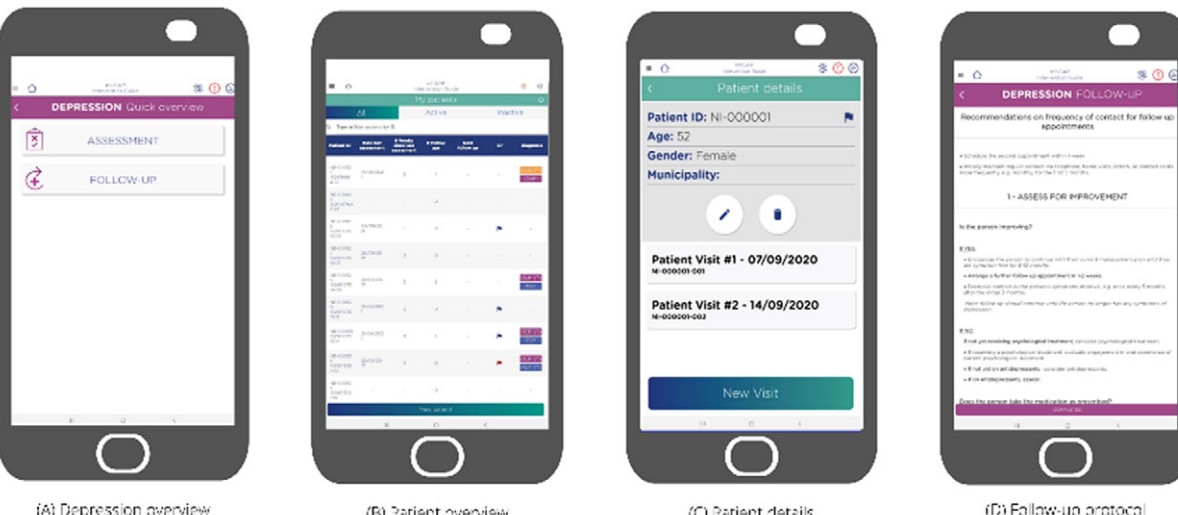

(A) Depression overview    (B) Patient overview    (C) Patient details    (D) Follow-up protocol

**Figure 3.** Follow-up care in the app prototype.

treatment adherence in the follow-up visits. They also recommended including the possible date for next follow-up visit in the application (Figure 3).

### Brief overview of the final app prototype

The detailed content of the final application and dashboard (i.e., patient data collected in the application as a part of EMILIA pilot) will be published separately. As indicated in Figure 1, the final application consists of four modules that are designed for specific purposes. As healthcare providers suggested, the home screen of the final application consists of 'Patients', followed by 'Master Chart' 'Assessment' and 'Management' modules. Broadly these four modules follow the features described above, with the patient icon providing access to a basic clinical record system, the master chart providing a summary of key mental health conditions, the assessment module providing the "yes/no" questions, primarily used for learning and providing management information. In the final app, there is a workflow that leads from registration (e.g., demographics) through the master chart, assessment and on to management for all selected conditions. Follow-up is also included under 'patients' module. A healthcare provider can operate the application only after creating a user account in the digital platform. It works in two modes 'patient mode' for assessing new patients and 'reference mode' for use as a learning tool.

### Discussion

This paper summarizes the development process and prototype testing of the mobile app-based WHO mhGAP-IG, and its potential functionality in the assessment and management mental health conditions in primary healthcare facilities in Nepal. The draft prototype was developed using a human-centered design approach among healthcare providers who were trained on mhGAP-IG and was based on earlier work which highlighted the importance of using a mobile health solution, as opposed to a solution that used a desktop computer. The results indicated that healthcare workers are more familiar with mobile applications than desktop applications. The prototype testing was conducted in three distinct rounds

by using role-plays with a standard case vignette, observations and semi-structured interviews. Healthcare providers experienced multiple technical and practical problems in the first round of the prototype testing. The prototype was rather slow which might have created problem in reviewing detailed symptoms of a particular mental health condition. This could have contributed to creating discrepancies between the assessment outcome generated by the prototype and the assessment outcome of healthcare providers. Many providers also reported a poor user experience with the prototype. This highlights the need to ensure that any digital device has a good user experience and is responsive to input to support the providers in conducting assessments. In addition, healthcare providers were overly engaged in the prototype (i.e., reading text, selecting reported symptoms), which affected the consultation, and highlighted the need for familiarity and training when using a digital device. Despite multiple technical and practical issues in the prototype, most of the healthcare providers were positive and motivated in using it. The key benefits of the app prototype reported by the healthcare providers were being reminded of the information they need to ask patients for assessment, and being provided a clear pathway in assessment, diagnosis, management and follow-up of mental disorders.

As some healthcare providers lacked experience in using smartphones, and their previous use of mobile phones was also limited to phone calls and some social media usage (Pokhrel et al., 2021), this might have contributed to some of the technical issues in the prototype such as difficulty selecting 'yes' or 'no' and using the "i" option which provides detailed symptoms of a particular mental health condition. Moreover, the technical issues in the prototype testing were not surprising in the context of slow internet connectivity in rural Nepal (Krishana, 2020). Some of the technical issues reported by healthcare providers in the prototype testing were also consistent with the studies conducted in other settings (Archibald et al., 2014; Bergquist et al., 2020). The common problems reported in testing mobile app-based intervention included small screen-size (Boruff and Storie, 2014); less user-friendliness (Seto et al., 2012), and lack of technical knowledge among healthcare providers (Broderick and Haque, 2015).

Other recommended functionality and features that could enhance motivation for use include incorporating a digital copy

of the mhGAP-IG, a quick walk-through video guide, and tutorials in the application. The participants were also optimistic that they could use the application without any trouble after having multiple rounds of practices in the prototype. Previous studies have also demonstrated that getting familiar with m-health is an important factor for its use (Putzer and Park, 2012; Hofstetter et al., 2013).

The key advantages of the mobile application reported by the healthcare providers were maintaining uniformity in the assessment and management of mental health conditions across primary healthcare facilities; verification of diagnosis and treatment; reviewing patients' data in the follow-up visits and using the patients' data in remote supervision. While the most frequently reported issue in the prototype was problems in selecting 'yes' or 'no' options, these were also the most liked feature by the participants. As these options not only aid in diagnosis but also help remind healthcare providers of important information to be asked during the assessment. These could be the possible reasons that participants liked these features. As the healthcare providers who participated in the study had multiple years of experience in using mhGAP-IG V2 protocols, this could be the reason they completed the role-plays (assessments and management using prototype) without any difficulties. Although healthcare providers endorsed the use of prototype in healthcare delivery, it is important to evaluate how this impacts the quality of the interpersonal interaction between providers and patients, the level of adequate detection and treatment initiation, and any potential benefit to the care and support provided to individuals. A pilot trial has been conducted to assess the feasibility, acceptability and appropriateness of this application in a real-world setting in a district in eastern Nepal (Taylor Salisbury et al., 2021b).

While healthcare providers demonstrated a high level of motivation and enthusiasm in using the mobile application (Pokhrel et al., 2021), some of the technical and practical problems that arose during the prototype testing suggest serious concerns in using the application in real-world settings. Some technical problems experienced by the healthcare providers in our study were also reported while using a digitally enabled integrated care system in rural Nepal (Citrin et al., 2018b). These problems might have occurred because of the lack of basic knowledge and technical skills among healthcare providers in using technology. This can be addressed through an additional training on the use of applications on smart mobile phones for those healthcare providers who lack experience in using smartphones. The disruption in the communication between healthcare providers and patients during consultation might have impact in maintaining quality of care and patients' acceptability of mobile technology-based services. This can be addressed by using the mobile application after completing the initial consultation with patients and may decrease when healthcare providers are experienced in using the application. A study conducted in United Kingdom reported that patients considered technology-based treatment positively thinking that use of technology can help clinicians to provide better care (Pinnock et al., 2006); therefore, patients may also prefer technology-based services if the use of technology improves quality of services, but likely only if some of the key issues related to its use and impact on the consultation are addressed. The findings indicate the need for applications to be responsive and take into account issues such as internet connectivity. The prototype software required a constant internet connection, whereas the final application to be tested in the pilot was designed for offline use. This suggests the importance of paying attention to any limitations in the technological infrastructure. However, as an internet connection is still required for initial login,

use of application in rural areas where internet connection is limited may be impacted. This problem can be minimized by using mobile data for log-in into application. In many cases, healthcare providers also used their mobile data to connect with the internet for log-in.

Our study and the final mobile application has several limitations. Although most of the healthcare providers suggested including a 'sending text messages' function in the app prototype to reduce dropout rate, the current application does not include this function, because it was not feasible within the scope of the current project (Pokhrel et al., 2021). 'Sending text messages' was reported to be effective in improving community health care and nutrition service utilization in rural Nepal (Acharya et al., 2020). If 'sending text message' function had been included in the final version, the app could have also provided a potential strategy to address the high dropout rate in mental health care in Nepal (Luitel et al., 2020). Second, the slowness of the prototype created many problems which affected interpretation of the observation and interview data. Third, the study was conducted among healthcare providers who were experienced in using the paper version of the mhGAP IG. The results could have been different if the study was conducted with healthcare providers without prior experience on mhGAP-IG. Lastly, the final application did not include all the features and functions that were recommended by healthcare providers. The healthcare providers were especially motivated if the e-mhGAP-IG could function as an electronic medical record (EMS) with contact information, follow-up scheduling, and other performance features for ongoing monitoring and evaluation. However, data security issues and the need for integration of government healthcare institutions may preclude development and use of e-mhGAP-IG as an EMS. Moreover, the lack of thorough documentation about physical health in the app may exacerbate issues of diagnostic overshadowing in which healthcare providers neglect physical health conditions once an individual is viewed as a 'mental health patient'.

While the application prototype has not been systematically evaluated yet, the findings of this study may have some implications to improve detection and management of mental disorders in primary healthcare facilities. First, healthcare providers reported that the prototype reminded them of the steps and processes to be followed in assessment and management of mental disorders. If the application is used systematically in consultation, this may help to improve detection of depression in primary care – as intended. While depression is one of the most prevalent mental health conditions among people attending primary healthcare facilities (Rait et al., 2009), trained primary healthcare providers detected only one-fourth of patients with depression right after the mhGAP-IG training in our previous study (Jordans et al., 2019). The mobile application could help to bridge this gap by increasing the detection rate of depression. Second, patients' data collected in the application can be reviewed by supervisors at a remote or distant location, and provide feedback immediately to the respective healthcare providers through phone calls or through other communication channels such as Viber, WhatsApp, *etc*. Third, the patients' data can also be linked directly to the Health Management Information System which helps to reduce paper work among primary healthcare providers. Finally, we used the human-centered design approach, a widely used methodology in co-creating digital interventions (IDEO, 2015), to understand the experiences and need of healthcare providers who were involved in delivery of mental health services in primary health care. This process identified the needs of health workers early on and highlighted a number of problems with

the existing prototype which could be rectified prior to developing the app. This shows the potential utility of this approach when developing complex digital applications for healthcare. By involving users in the development of the application, we have a better chance of developing and testing something that healthcare providers view as appropriate, feasible and acceptable in using in the routine healthcare system in Nepal. This may increase the trust and ownership among healthcare providers.

## Conclusion

In this paper, we have described the systematic process that was followed in development and testing the mobile app-based clinical guideline supporting the integration of mental health in primary healthcare settings (WHO e-mhGAP-IG). Despite some technical and practical problems in the initial version of the prototype, healthcare providers perceived it feasible, acceptable and appropriate in using it in the real-world settings. This suggests the potential utility of decision support tools, yet a rigorous outcome evaluation of the final application needs to be performed in the health facilities located in remote areas to assess the feasibility of implementing the application in the areas where internet connection and other infrastructure are not well developed.

**Open peer review.** To view the open peer review materials for this article, please visit http://doi.org/10.1017/gmh.2023.69.

**Data availability statement.** Interested parties may notify the EMILIA investigators of their interest in collaboration, including access to the data-set analyzed here, through the following email: luitelnp@gmail.com.

**Acknowledgements.** We want to thank Mr. Damodar Rimal, Ms. Sirjana Pandey, Ms. Rojina Karmacharya and Mr. Manoj Dhakal for their support in the data collection. The authors alone are responsible for the views expressed in this article and they do not necessarily represent the views, decisions or policies of the institutions with which they are affiliated. N.P.L. is supported by National Institute for Health Research (NIHR) and Wellcome Trust under the NIHR-Wellcome Partnership for Global Health Research (grant reference 222001/Z/20/Z). G.T. is supported by the National Institute for Health and Care Research (NIHR) Applied Research Collaboration South London (NIHR ARC South London) at King's College Hospital NHS Foundation Trust. The authors alone are responsible for the views expressed in this article, and they do not necessarily represent the views, decisions, or policies of the institutions with which they are affiliated, including the World Health Organization, NIHR or the Department of Health and Social Care. G.T. is also supported by the UK Medical Research Council (UKRI) for the Indigo Partnership (MR/R023697/1) awards. For the purpose of open access, the author has applied a Creative Commons Attribution (CC BY) license (where permitted by UKRI, 'Open Government License' or 'Creative Commons Attribution No-derivatives (CC BY-ND) license' may be stated instead) to any Author Accepted Manuscript version arising.

**Author contribution.** N.P.L., T.T.S., M.J., B.A.K., G.T., and K.C. were responsible for the study design. N.P.L. and K.C. were responsible for supervision of data collection. N.P.L. and P.P. were responsible for data collection. N.P.L., K.P. and P.P. drafted the first version of the manuscript; all authors reviewed and revised the manuscript. All authors read and approved the final manuscript.

**Financial support.** This study is funded by the UK Medical Research Council in relation to the Emilia Project (MR/S001255/1). The funders had no role in study design, data collection and analysis, and decision to publish or preparation of the manuscript. The authors had full control of all primary data.

**Competing interest.** The authors declare none.

**Ethical standard.** The study received ethical approval from the Nepal Health Research Council (ref: 810/2018), Kings College London Research Ethics Committee (ref: LRS-18/19–8,358) and the World Health Organization Research Ethics Review Committee (ref: ERC.0003246).

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
