## [Reviewer Report]

29 May 2023

Kathmandu, Nepal

To: The Editor-in-Chief, 

Cambridge Prisms: Global Mental Health

Re: Submission of manuscript “Development and functioning of the mobile app-based mh-GAP Intervention Guide in detection and treatment of people with mental health conditions in primary healthcare settings in Nepal”

Dear Editor-in-Chief,

Please consider the accompanying paper for publication in Cambridge Prisms: Global Mental Health. 

There is growing global need for scalable approaches to training and supervising primary care workers to deliver mental health services. In recent years, mobile technologies have increasingly been used in health care, particularly in educating healthcare workers, supporting clinical care, increasing adherence and recovery, and in remote supervision. None have attempted to develop a clinical decision support tool based on mhGAP. We developed a mobile app-based clinical guideline which uses the WHO mhGAP-IG-V2 protocol. The content of the mobile app was developed in both Nepali and English languages. The app provides provisional diagnoses and treatment options based on reported symptoms. Health workers found the app prototype useful in reminding them of the process of assessment and management of mental disorders. The initial feasbility work suggests that the mhGAP mobile app could be a useful method to support primary healthcare providers in improving detection of people with mental health conditions and initiation of evidence-based treatment in primary health care facilities. 

We warrant that the information contained in the manuscript represents original work and is not under consideration for publication elsewhere. No competing interests need to be disclosed for any of the authors.

Yours sincerely,

Nagendra P. Luitel (nagendra.luitel@email.gwu.edu)

Kriti Pudasaini

Pooja Pokhrel

Bishnu Lamichhane

Kamal Gautam

Sandarba Adhikari

Akerke Makhmud

Tatiana Taylor Salisbury

Nicole Votruba

Eric Green

Neerja Chowdhary

Mark JD Jordans

Brandon A. Kohrt

Tarun Dua

Graham Thornicroft

Kenneth Carswell

---

## [Reviewer Report]

Congratulations to the authors on the paper and development of the mhealth tool. I have minor comments listed below.

1. In the impact statement, the authors note that “health workers were able to provide minimally adequate treatment as per mhGAP protocol”- I am not sure what minimal adequate treatment means. I would recommend adding some objectivity, even an example, to minimize subjectivity biases. I see that the minimally adequate treatment is also alluded to in the introduction section, and a description would be helpful. Page 2, line 14

2. For a non-GMH audience or even a non-western population, it may not be clear why people with depression and common mental health problems are likely to access services at primary care. I understand this is more obvious to a GMH audience, but since these claims are well-established in the literature, adding citations would increase credibility. Page 4, lines 18-19

3. I wonder if Page 4, lines 50-55, is missing something. From talking about digital platforms, the authors jump to issues with a reliable connection that seems slightly off.

4. Citations: in the intro paragraph. For example, see page 5, lines 18-19 and 20-21.

5. Is there a reason the authors manually coded and analyzed the qualitative data for a relatively large number of participants?

6. Is that supposed to be the figure description? Page 8, lines 4-5?

7. Unclear sentence, page 8, lines 11-15

8. Complex sentence structure pg. 8, lines 41-42

9. Why did the device freeze? Is this because of the app or because the providers had machines that were freezing more? Page 8, line 50

10. I am unclear how the speed of working is associated with the discrepancy in suspected and actual diagnosis. This may need to be clarified. Page 9, lines 3-5.

11. While I understand the need for a nonidentifiable record number for each person, I am unsure why the medication module was removed. Pg 9, lines 25-31.

More general comments:

The authors have employed multiple stages and sub-stages within the HCD process. Practically, I can see how this was helpful in app development, but theoretically, it was a bit hard to keep track of the process used. I would recommend a visual depiction of the process that outlines the various states and sub-stages of development. A simple flow diagram or some visual representation would be helpful.

The healthcare providers in this study were experienced users of mhGAP. How experienced were they? Can a range of training and experience years be added to elaborate the claim?

Check abbreviation: EMS is spelled out as an electronic medical record. Pg 13, lines 16-17

The discussion section on using mHealth tool seems more relevant for a community-based setting, but the study was conducted at a clinic. I can speculate why, but it may be helpful to give more context about why a mhealth (not EHR, computer-based) mhGAP was adopted. And what value does the mobile-based mhGAP add to service delivery that a traditional EHR may not?

---

## [Reviewer Report]

This paper addresses a very important issue. Trained and sufficient workforce for managing common mental disorders is a huge problem in LMICs. Mobile and Web-based applications could be a solution to this problem. I appreciate the authors for undertaking this research.

Paper describes development of a mobile based application to assist healthcare workers in diagnosis and management of (selected) mental health problems. Authors claim that their methodology is based on HCD but do not explain the phases in such a detail that readers can understand the process of development.

At many points, authors do not present sufficient data to support their claims. For example, methods state that qualitative interviews were conducted and thematic analysis was performed but the results of thematic analysis are seen nowhere in the manuscript. Also authors have not provided any demographic characteristics of the study sample. Were they both males and females, what age group/experience etc. Especially in discussion section, many new findings are introduced that have not been mentioned in results previously. For example

“While healthcare providers demonstrated a high level of motivation and enthusiasm in using mobile application, some of the technical and practical problems that arose during the prototype testing suggest serious concerns in using the application in real-world settings. Some technical problems experienced by the healthcare providers in our study were also reported while using digitally-enabled integrated care system in rural Nepal”

What were these technical and practical problems? This is very crucial information for developers, researchers and end-users to consider while doing/using similar apps in similar context. I would suggest to make this a separate section in findings of prototype testing.

Was it off-line app or needed internet connectivity to operate? How stable is the internet connection in the study area? Was that a problem for users?

How did clients react to app usage by health care providers? Is there any data on that? If yes, should be included in findings. This could just be observations/experiences of users (healthcare providers).

Authors at many places in manuscript have referred to other paper published or yet to be published, which is okay as far as it does not withhold any information necessary to understand or contextualize findings of this paper. Please include the necessary details